# Characterization of a Temporal Profile of Biomarkers as an Index for Ischemic Stroke Onset Definition

**DOI:** 10.3390/jcm10143136

**Published:** 2021-07-15

**Authors:** Andrés Da Silva-Candal, Antonio Dopico-López, Maria Pérez-Mato, Manuel Rodríguez-Yáñez, José M. Pumar, Paulo Ávila-Gómez, José Castillo, Tomás Sobrino, Francisco Campos, Pablo Hervella, Ramón Iglesias-Rey

**Affiliations:** 1Clinical Neurosciences Research Laboratories (LINC), Health Research Institute of Santiago de Compostela (IDIS), 15006 Santiago de Compostela, Spain; adopicolopez@gmail.com (A.D.-L.); pauloavilagomez@gmail.com (P.Á.-G.); jose.castillo.sanchez@sergas.es (J.C.); Tomas.Sobrino.Moreiras@sergas.es (T.S.); francisco.campos.perez@sergas.es (F.C.); 2Neurovascular Diseases Laboratory, Neurology Service, University Hospital Complex of A Coruña, Biomedical Research Institute (INIBIC), 15706 A Coruña, Spain; 3Neuroscience and Cerebrovascular Research Laboratory, Department of Neurology and Stroke Center, La Paz University Hospital, Neuroscience Area of IdiPAZ Health Research Institute, Universidad Autónoma de Madrid, 28046 Madrid, Spain; maria.perez.mato@idipaz.es; 4Stroke Unit, Department of Neurology, Hospital Clínico Universitario, 15006 Santiago de Compostela, Spain; manyanez@yahoo.es; 5Department of Neuroradiology, Hospital Clínico Universitario, Health Research Institute of Santiago de Compostela (IDIS), 15006 Santiago de Compostela, Spain; josemanuel.pumar@usc.es

**Keywords:** stroke, biomarkers, glutamate, neuroprotection, inflammation

## Abstract

Background and purpose: Stroke is a dynamic process in terms of molecular mechanisms, with prominent glutamate-mediated excitotoxicity at the onset of symptoms followed by IL-6-mediated inflammation. Our aim was to study a serum glutamate/IL-6 ratio as an index for stroke onset definition. Methods: A total of 4408 ischemic stroke patients were recruited and then subdivided into four quartiles according to latency time in minutes (0–121, 121–185, 185–277 and >277). Latency time is defined as the time between stroke onset and treatment at the neurological unit. The primary endpoint of the study was the association of early latency times with different clinical aspects and serum markers. Serum glutamate and interleukin-6 (IL-6) levels at admission were selected as the main markers for excitotoxicity and inflammation, respectively. Results: Glutamate serum levels were significantly higher in the earlier latency time compared with the higher latency times (*p* < 0.0001). IL-6 levels were lower in early latency times (*p* < 0.0001). Patients with a glutamate/IL-6 index on admission of >5 were associated with a latency time of <121 min from the onset of symptoms with a sensitivity of 88% and a specificity of 80%. Conclusions: The glutamate/IL-6 index allows the development of a ratio for an easy, non-invasive early identification of the onset of ischemic stroke symptoms, thus offering a new tool for selecting early stroke patient candidates for reperfusion therapies.

## 1. Introduction

“Time is brain” is one of the mantras of stroke treatment in clinical practice. In an ischemic stroke, a typical large vessel occlusion causes the destruction of around 1.9 million neurons, 14 billion synapses and 12 km of myelinated fibers in a single minute [1]. Recanalization of the occluded vessel remains the priority in clinical efforts and undoubtedly the strongest dependence exists between time and a good outcome [2,3,4,5,6]. Underlying the primary mechanisms that lead to a stroke, there are a series of sub-processes such as inflammation [7] or excitotoxicity [8] which, despite their importance, are not currently treated in the clinical practice. They could be the main target of experimental neuroprotective treatments [9,10,11,12] to potentially improve patient outcome. The time profile should therefore be thoroughly reviewed so that the focus is on patient-specific efforts that are provided at the appropriate time frame.

Stroke biomarkers have been widely studied [13,14,15], mostly as predictors for severity [16,17], diagnosis [18,19], risk [20], hemorrhagic transformation [20,21] or outcome [22,23]. However, stroke is a pathology in which the onset of symptoms is a key factor for the establishment of clinical treatments. For this reason, it is extremely necessary to define reliable markers to approximate the onset of symptoms. 

In this study we analyzed two time-dependent processes of ischemic stroke. Firstly, we examined glutamate levels as a marker of excitotoxicity [24], a molecule involved in secondary neuronal death during the acute phase and related to the severity of injury and patient outcome [25]. Secondly, we analyzed interleukin-6 (IL-6) [16,26], a pro-inflammatory molecule implicated in the early stages of stroke [27,28] which has also been shown to be involved in different pathways that increase the severity of damage [29] and patient outcome [28]. We carried out a combined analysis of these two temporal markers in order to develop an index to determine stroke symptoms onset.

## 2. Materials and Methods

### 2.1. Primary Endpoint

As stroke is a dynamic process in terms of molecular mechanisms, with prominent glutamate-mediated excitotoxicity at the onset of symptoms followed by IL-6-mediated inflammation, our aim was to study a serum glutamate/IL-6 ratio as an index for stroke onset definition.

### 2.2. Study Design

This is a retrospective study conducted on a prospective registry of patients with acute cerebrovascular disease consecutively admitted to the Stroke Unit of the University Clinical Hospital of Santiago de Compostela (Spain) between January 2008 and December 2017. From January 2008, we prospectively included all ischemic stroke patients admitted to the Stroke Unit of the University Clinical Hospital of Santiago de Compostela in the BICHUS registry.

### 2.3. Population of Study and Clinical Variables

From January 2008 to December 2017 a total of 4775 ischemic stroke patients were recruited for this study. Of these, a total of 400 patients suffered awakening strokes. A total of 377 were excluded from the analysis due to missing data regarding latency times (157 patients) or lost to follow up at 3 months (220 patients). Latency time is defined as the time (in minutes) between the onset of symptoms and arrival at hospital. As far as the awakening stroke is concerned, latency time is defined as the time frame between the last time the patient showed no evidence of symptoms and arrival at hospital. Patients were subdivided into quartiles on the basis of this latency. Quartile 1 (Q1) consisted of 1103 patients with a latency time between 0–121 min, Quartile 2 (Q2) included 1103 patients with a latency time between 121–185 min, Quartile 3 (Q3) comprised 1117 patients with a latency time between 185–277 min and Quartile 4 (Q4) included 1085 patients with a latency time higher than 277 min. Demographic data and clinical variables were analyzed for each quartile of latency time. The patients’ ages and medical histories of comorbidities were included in this analysis. 

The hemorrhagic transformation of ischemic infarction was evaluated using the European Cooperative Acute Stroke Study III (ECASS III) criteria [22]. Stroke severity was measured by the National Institutes of Health Stroke Scale (NIHSS), which was performed by trained neurologists of the Stroke Unit. Early neurological improvement was defined as a decrease of ≥8 points in the NIHSS in the first 24 h and early neurological deterioration as an increase of ≥4 points in the first 48 h. The etiological diagnosis of IS was performed according to TOAST criteria [30].

Serum markers were analyzed at admission, using consecutively registered patients according to predetermined inclusion criteria. As a marker for excitotoxicity, glutamate serum levels were analyzed (*n* = 1568 total). IL-6 (*n* = 1430) was selected as an inflammation marker. 

### 2.4. Analytical Measurements

Biochemistry, hematology and coagulation tests were assessed at the hospital’s central laboratory. For the molecular measurements, venous blood samples were collected in Vacutainer tubes (Becton Dickinson, San Jose, CA, USA) on admission, and 24 ± 12 h and/or 48 ± 12 h from stroke onset. After clotting for 60 min, blood samples were centrifuged at 3000× *g* for 10 min, and the serum was immediately aliquoted, frozen and stored at −80°C until analysis.

Glucose levels, glycosylated hemoglobin, leukocytes, red blood cells, platelets, fibrinogen, C-reactive protein, total and fractionated cholesterol, triglycerides, NT-proBNP, vitamin D and cholecalciferol were measured on admission. Serum levels of glutamate were measured on admission and after 24 h. Serum glutamate concentration was measured by high-performance liquid chromatography (1260 Infinity II; Agilent Technologies, Santa Clara, CA) using the AccQ-Tag precolumn derivatization method for amino acid analysis (Waters, Milford, MA, USA) following the method described elsewhere [31]. Serum levels of IL-6 were also measured on admission and after 24 h using an immunodiagnostic IMMULITE 1000 System (Diagnostic Products Corporation, Los Angeles, CA, USA). 

To calculate the Glut/IL-6 ratio, serum glutamate values were divided by serum IL-6 values. Of the total series, the number of patients with glutamate levels analyzed was 1568, and 1430 patients were analyzed for the IL-6 levels. A total of 1221 patients had both glutamate and IL-6 levels. The univariate, multivariate analyses and the Glut/IL-6 ratio described in the results were performed on these patients. A flowchart showing the study design of an analysis has been included as Appendix A

### 2.5. Neuroimaging Studies

All ischemic stroke patients included in the study underwent cerebral computed tomography (CT) on admission and between days 4 and 7 after stroke onset. Besides, multimodal magnetic resonance imaging (MRI) was also performed on admission in those ischemic stroke patients who were candidates to reperfusion treatments. Lesion volumes were measured using ABC/2 method [32] until 2016 and from then on through automated planimetric method in diffusion-weighted imaging (DWI)-MRI on admission and between the 4th and 7th day in ischemic stroke patients. From 2012, an MRI angiography was performed in all patients who were candidates for reperfusion treatments. All neuroimaging studies were performed by neuroradiologists blinded to clinical and analytical data. The infarct was calculated as the volume difference between the first DWI and the second imaging study (CT between the 4th and the 7th day and 24 h, respectively). 

### 2.6. Statistical Analysis

An initial descriptive analysis was performed. Significant variables were added to a multivariate regression model. Results were defined as percentages for categorical variables and as mean ± standard deviation (SD) or median and range (25th–75th percentiles) for continuous variables depending on whether their distribution was normal or not. Kolmogorov–Smirnov test was applied to assess normality. Then, statistical inference was carried out with the chi-square test, ANOVA or H of Kruskal–Wallis according to the nature of the contrast variable and its adjustment to normality. Bivariate correlations were performed using Pearson’s or Spearman coefficients. Cut-off points were calculated in the variables of interest with a COX regression. Results were shown as odds ratios (ORs) with 95% confidence intervals (CI 95%) after a logistic regression analysis where latency time was defined as the dependent variable. *p*-value < 0.05 was considered to be statistically significant in all tests. The statistical analyses were conducted in SPSS 21.0 (IBM, Armonk, NY, USA).

## 3. Results

### 3.1. Time Dependence of Early Latency Times and Demographic Aspects 

In the univariate analysis, the age of the patients when treated for stroke (*p* = 0.026), the percentage of female patients (*p* < 0.0001), the percentage of previous transient ischemic attack (TIA) (*p* = 0.027), the duration of the TIAs (*p* = 0.040) and the percentage of anticoagulated patients (*p* = 0.08) were found to vary significantly in the different quartiles (Appendix A). We compared the shortest latency time of 0–121 min (Q1) with the rest of the quartiles in a multivariate analysis. The adjusted model showed that early latency times were associated with two demographic variables: directly with the year of recruitment (OR: 1.13; CI 95%: 1.09–1.16; *p* < 0.0001), and inversely with being female (OR 0.29; CI 95% 0.24–0.35; *p* < 0.0001) (Table 1).

### 3.2. Time Dependency of Clinical Aspects

In order to evaluate whether clinical variables and prognosis might be influenced by different latency times, a bivariate analysis of different clinical and prognosis-related aspects of patients distributed among the different latency time ranges was performed (Table 2) using a multivariate regression model that compared the lower latency time range (Q1) with the rest of the quartiles (Q2, Q3, Q4) (Table 3). In the non-adjusted model, inflammation markers such as leukocytes (OR: 0.95; CI 95%: 0.93–0.98 *p* < 0.0001) and C-reactive protein (OR: 0.91; 0.96–1.00; CI 95%: *p* < 0.0001) were inversely correlated with early latency times. Other factors such as receiving fibrinolytic treatment (OR: 1.16; CI 95%: 0.97–1.37 *p* = 0.009) or mechanical thrombectomy (OR: 1.15; CI 95%: 1.08–2.12 *p* = 0.016) were more related to early latency times. Conversely, those patients who received early hospital care were inversely related to a poor outcome after 3 months (OR: 0.58; CI 95%: 0.50–0.68; *p* < 0.0001). However, the significance of all these associations was lost after adjusting for confounding factors.

### 3.3. Time Dependence of Circulating Biomarkers 

Glutamate serum levels were elevated in latency times under 121 min (Appendix A), matching with the acute stage of glutamatergic excitotoxicity in the brain (Figure 1a). Meanwhile, IL-6 remained lower in Q1 than in higher latency times (*p* < 0.0001) (Figure 1b). Glutamate and IL-6 levels showed a similar profile when awakening strokes were suppressed from the analysis (Appendix A).

The clear difference observed in serum levels of glutamate and IL-6 with the passing of time in the different quartiles suggests that they can be used as biomarkers to establish a time range in which stroke symptoms began. Individual ROC curves showed that a glutamate concentration on admission of >150 µM was associated with a latency time of <121 min from the onset of symptoms, with a sensitivity of 90.03% and with a specificity of 70.3% (Figure 2a). Conversely, an IL-6 concentration on admission <18 pg/mL was associated with a latency time of <121 min, with a sensitivity of 88% and with a specificity of 76% (Figure 2b).

### 3.4. Excitotoxicity/Inflammation Index as a Time Marker for Early Symptom Onset

Glutamate and IL-6 showed an opposite profile in Q1 suggesting that the combination of both markers may be a strong indicator of early symptoms. Based on this, we analyzed the Glutamate/IL-6 ratio in a univariate analysis for the different quartiles (ANOVA test *p* < 0.0001) showing a strong relationship with early latency times (Figure 2c). In light of the differences found, we analyzed its relevance as a discriminative factor through the elaboration of an ROC curve in the Q1 patients. The results showed that those patients with a Glutamate/IL-6 index on admission of >5 were associated with a latency time < 121 min from the onset of symptoms with a sensitivity of 88% and with a specificity of 80% (Figure 2d).

## 4. Discussion

Time is one of the main variables related to patient prognosis in ischemic stroke. Although many studies have focused on the time of arrival and the possibility of treatment, their conclusions are disparate [33] as they were performed under different contexts, aimed at different objectives and used different methodologies and perspectives. Throughout this study, we analyzed how early latency influences different demographics, prognosis and molecular serum markers in ischemic patients. In our paper, we focused on periods of latency of the acute stage of stroke in order to evaluate the time profile for the use of future neuroprotective treatments, and also for establishing the basis for a quick and non-invasive determination of the time frame for the onset of symptoms. This is especially necessary in the case of “awaking” strokes. In many cases, the last time the patient was conscious is taken as the reference for thrombolytic treatment, which seriously limits clinical practice.

For this analysis we have focused on the shortest latency time range of 0–121 min (Q1), which is considered a very early time of care, to better fine-tune the evaluation of the effect on the variables. Moreover, this time frame is optimal for the administration of thrombolytic treatments.

As for demographic data, the year in which patients were treated was associated with early latency times, which translated into more patients treated as compared to earlier years. The rapid identification of symptoms by patients as well as a progressive improvement in stroke units over the years are probably the causes of the decrease in latency times, allowing acute care for patients with ischemic stroke. We found a significantly lower number of female patients in Q1. Contrary to what other studies suggest, this may be attributed to social phenomena, with no sex-related differences in terms of time of arrival [33,34].

In the multivariate adjusted models of our study, we did not see dependence between any clinical variable and latency times under 121 min. We emphasize that receiving fibrinolytic treatment or the performance of thrombectomy were not associated with latency times under 121 min, probably because the time ranges of Q1, Q2 and Q3 (all under 277 min) are all time ranges within the therapeutic window of thrombolytic treatments. 

Finally, we analyzed two main serum parameters of stroke patients. Glutamate is considered one of the gold standards in excitotoxicity processes [35] and the target of several neuroprotection studies in ischemic stroke [12]. For this reason, we considered it as a suitable marker for excitotoxicity. Although the glutamatergic window is considered wider [4], in our study we found that those patients with latency times lower than 121 min showed higher glutamate levels than other time ranges, displaying a time-dependent nature. It is important to note that increases in the systemic levels of glutamate do not fully represent variations at brain level, since glutamate is a proteinogenic amino acid that participates in numerous biological reactions. Therefore, measuring just plasma levels may be masking smaller ups and downs. However, our study suggests that new neuroprotective treatments against glutamatergic excitotoxicity should be administered within 2 h after the onset of symptoms. 

On the other hand, we analyzed the levels of IL-6, an inflammatory biomarker in stroke associated with poor prognosis [36]. IL-6 levels showed a significant difference between patients in Q1 and the remaining quartiles, being higher in the latter, showing a profile similar to other studies [7]. Based on our study results, the data reliably suggests that no major benefit is obtained by an early administration (within 2 h) of an anti-inflammatory treatment.

Due to their time dependent nature, the opposite profile of these two markers and the meticulous analysis of the latency time, we consider the ratio between the two as an optimal predictor of the onset of symptoms as shown in the ROC curve. The glutamate/IL-6 ratio showed a remarkable specificity and sensitivity, higher than 95%, with the capacity to identify patients with a latency time of 0–2 h, which could prove a useful tool for developing a marker to identify patients within the time frame for reperfusion treatments, including patients with a wake-up stroke. 

The main limitation of this study is its retrospective nature, as unfortunately serum biomarkers were not analyzed for all patients as discussed elsewhere in this paper, which detracts from the statistical power of the results. In addition, further prospective studies with the elaboration of a marker profile in patient populations stratified in a concrete type of stroke and protocolized blood collection times would allow us to obtain more specific and possibly more precise temporal data. 

Time has always been one of the main paradigms in stroke and has been a key to knowing whether a patient was going to be treated. In this regard, the clinical implications of developing a ratio that leads to an early identification of the onset of symptoms in a non-invasive and easy way are extremely exciting as the reperfusion or surgical treatment could be expanded to those cases in which the onset time is unknown. This would translate into better clinical practice and improved patient outcome.

## 5. Conclusions

Ischemic stroke is marked by a strong temporal dependence. The impossibility of determining the onset of the symptoms has always been a huge limitation when applying pharmacological fibrinolytic treatment in the clinical practice. 

To address this problem, throughout this study two serum markers, glutamate and IL-6, were analyzed in patients with different latency times. Both markers showed an opposite temporal profile with special relevance in short latency times (between 0–121 min) in which an increase in glutamate levels was observed compared to the rest of the latency times; conversely, IL-6 levels remained low compared to the rest of the latency periods, in which a significant increase was observed.

The combination of these two markers resulted in the development a Glutamate/IL-6 index. This ratio enabled us to determine the onset of ischemic stroke symptoms. Its low invasiveness and quick measurement make this strategy a new tool for selecting early stroke patient candidates for reperfusion therapies.

## Figures and Tables

**Figure 1 jcm-10-03136-f001:**
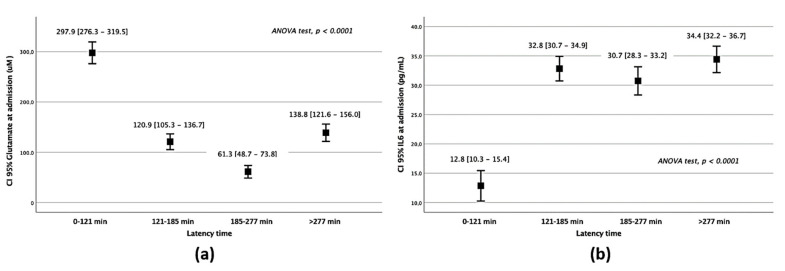
Serum marker analysis of patients belonging to different latency times. Glutamate (**a**) and IL–6 (**b**) serum levels on admission regarding the latency times of attendance.

**Figure 2 jcm-10-03136-f002:**
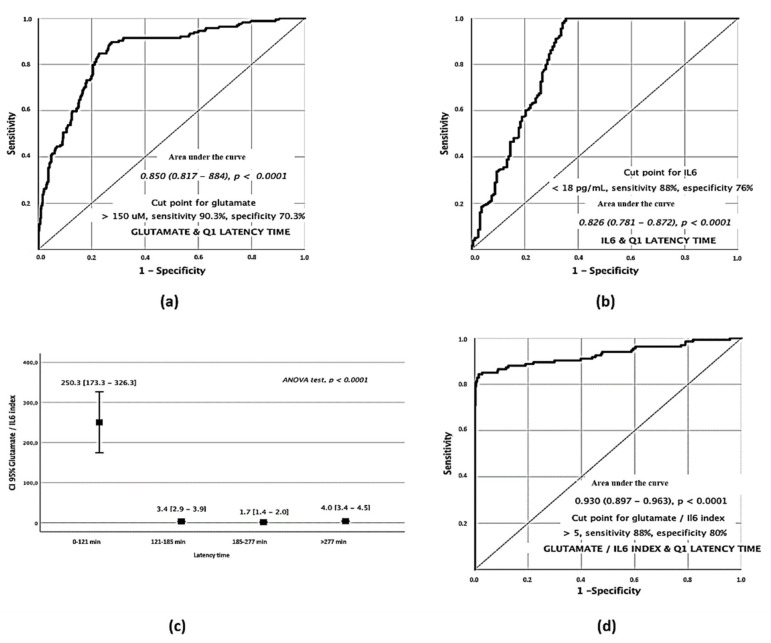
Serum biomarkers analysis of stroke patients on admission. Glutamate ROC curve for Q1 latency times (**a**). IL–6 ROC curve for Q1 latency times (**b**). ANOVA test of Glutamate/IL–6 index in the different quartiles (**c**). ROC curve for the Q1 Glut/IL–6 index (**d**).

**Table 1 jcm-10-03136-t001:** Multivariate analysis of demographic aspects using latency times of 0–121 min (Q1) as dependent variable.

	Not Adjusted			Adjusted		
	OR	CI 95%	*p*	OR	CI 95%	*p*
Year *	1.12	1.09–1.15	<0.0001	1.13	1.09–1.16	<0.0001
Age	0.99	0.98–0.99	<0.0001	0.99	0.99–1.01	0.683
Woman	0.30	0.24–0.34	<0.0001	0.29	0.24–0.35	<0.0001
Hypertension	0.82	0.71–0.96	0.014	0.96	0.77–1.08	0.296
Smoking	1.45	1.19–1.76	<0.0001	1.00	0.81–1.26	0.944
Enolism	1.61	1.29–2.01	<0.0001	1.10	0.87–1.40	0.429

Note: Year * = Year of admission to hospital center, OR = Odds Ratio, CI = Confidence Interval.

**Table 2 jcm-10-03136-t002:** Univariate analysis of clinical aspects using latency times as dependent variable.

	Q1	Q2	Q3	Q4	*p*
Temperature at admission, °C	36.3 ± 0.6	36.3 ± 0.6	36.4 ± 0.6	36.4 ± 0.7	0.054
Glycemia, mg/dL	137.3 ± 60.1	135.9 ± 53.6	138.7 ± 56.3	138.1 ± 61.3	0.793
Leukocytes, ×10^3^/mmc	8.8 ± 3.1	9.1 ± 3.2	9.4 ± 3.2	9.2 ± 3.1	<0.0001
Fibrinogen, mg/dL	434.7 ± 100.4	439.7 ± 107.6	449.8 ± 108.7	453.9 ± 95.4	0.002
Microalbuminuria, mg/24 h	8.0 ± 31.9	5.5 ± 23.7	6.8 ± 30.6	4.9 ± 23.9	0.337
C reactive protein, mg/L	2.7 ± 3.5	3.2 ± 3.9	4.2 ± 4.7	3.3 ± 3.9	<0.0001
Glycosylated haemoglobin, %	6.3 ± 4.7	6.1 ± 1.1	6.2 ± 1.3	6.2 ± 1.4	0.403
LDL cholesterol, mg/dL	108.3 ± 38.1	110.1 ± 37.3	106.2 ± 36.9	110.6 ± 37.4	0.258
HDL cholesterol, mg/dL	40.0 ± 14.4	45.1 ± 28.3	42.0 ± 16.0	42.4 ± 18.1	0.001
Triglycerides, mg/dL	119.9 ± 71.9	115.7 ± 53.8	122.8 ± 64.1	123.0 ± 67.0	0.161
Sedimentation rate, mm/hr	24.2 ± 22.6	25.7 ± 23.8	26.6 ± 22.9	27.5 ± 23.6	0.067
ProBNP, pg/mL	1469.9 ± 1784.1	1703.2 ± 1804.5	1685.8 ± 1834.7	1608.8 ± 1973.1	0.128
Vitamin D, ng/mL	19.9 ± 9.8	19.3 ± 9.9	18.1 ± 8.3	18.8 ± 9.3	0.145
Intima-media thickness, mm	0.8 ± 0.2	0.9 ± 0.9	0.9 ± 0.7	1.0 ± 0.9	0.215
Fibrinolytic treatment, %	27.1	39.1	31.9	2.0	<0.0001
Thrombectomy, %	5.9	5.1	5.3	1.5	<0.0001
NIHSS at admission	14 (8, 20)	14 (9, 20)	14 (9, 19)	12 (7, 18)	<0.0001
NIHSS at 24 hours	6 (2, 14)	8 (3, 15)	10 (5, 17)	6 (2, 14)	<0.0001
NIHSS at 48 hours	5 (1, 12)	6 (2, 14)	8 (2, 16)	6 (2, 14)	<0.0001
NIHSS at discharge	6 (2,11)	47 (2, 12)	8 (2, 13)	7 (2, 12)	<0.0001
Early neurological improvement, %	35.1	28.8	19.2	16.9	<0.0001
Early neurological deterioration, %	17.2	25.0	40.7	17.2	<0.0001
TOAST:					0.102
Atherothrombotic, %	22.9	22.1	25.0	24.8	
Cardioembolic, %	35.1	40.1	38.7	36.0	
Lacunar, %	9.6	5.4	5.5	10.1	
Indeterminate/others, %	32.4	32.6	30.9	29.5	
Ischemic DWI volume at admission, mL	64.2 ± 125.2	38.7 ± 54.6	32.4 ± 39.9	39.9 ± 105.8	0.148
Clinical-DWI mismatch, %	18.0	16.9	9.1	8.5	0.001
Infarct volume (CT 4th–7th day), mL	47.8 ± 84.6	50.2 ± 80.3	66.6 ± 92.7	56.4 ± 86.6	<0.0001
Growth of the infarct, mL	7.6 ± 19.4	1.9 ± 25.1	5.9 ± 13.0	9.7 ± 21.7	0.165
Haemorrhagic transformation, %	14.0	12.6	10.3		0.048

Note: LDL = Low-Density Lipoprotein, HDL = High-Density Lipoprotein, DWI = Diffusion Weighted Image.

**Table 3 jcm-10-03136-t003:** Multivariate analysis adjusted and non-adjusted of the clinical aspects comparing Q1 with Q2, Q3 and Q4 latency times.

	Not Adjusted			Adjusted		
	OR	CI 95%	*p*	OR	CI 95%	*p*
Leukocytes	0.95	0.93–0.98	<0.0001	0.92	0.82–1.04	0.176
Fibrinogen	0.99	0.99–1.00	0.005	0.99	0.99–1.01	0.527
C reactive protein	0.94	0.91–0.96	<0.0001	0.94	0.83–1.06	0.344
HDL-cholesterol	0.99	0.98–0.99	0.001	0.98	0.96–1.01	0.144
Fibrinolytic treatment	1.16	0.97–1.37	0.009	1.74	0.59–5.17	0.316
Thrombectomy	1.15	1.08–2.12	0.016	0.16	0.02–1.12	0.065
Early neurological improvement	1.98	1.67–2.36	<0.0001	0.53	0.09–2.93	0.046
Early neurological deterioration	0.60	0.41–0.87	0.007	1.62	0.27–9.81	0.599
Clinical-DWI mismatch	1.73	1.18–2.52	0.005	4.23	0.93–19.13	0.061
NIHSS at admission	0.98	0.97–0.99	0.016	1.02	0.95–1.10	0.537
Infarct volume	0.99	0.99–1.00	0.011	0.99	0.97–1.01	0.420
Hemorrhagic transformation	1.24	1.00–1.28	0.035	1.37	0.33–5.74	0.667

Note: NIHSS = National Institutes of Health Stroke Scale, HDL = High-Density Lipoprotein DWI = Diffusion-Weighted Imaging.

## Data Availability

The datasets generated during and/or analyzed during the current study are available from the corresponding author on reasonable request.

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
