# Peer review of "Characterization of a Temporal Profile of Biomarkers as an Index for Ischemic Stroke Onset Definition"

_jcm, 2021, doi:10.3390/jcm10143136_

Round 1

Reviewer 1 Report

This paper reports that glutamate/IL-6 index on admission >5 is associated with latency time of < 121 minutes from onset of symptoms. This study included sufficient number to analyze (4775 ischemic stroke patient), but I think there are some points that authors had better revise. At first, a lot of research papers about biomarkers of stroke onset has been reported. Authors had better write these data more in detail.  Secondly, authors selected two inflammation biomarkers, glutamate and IL-6. Authors had better write why they chose only these 2 markers in detail because there must be other inflammation biomarkers to became candidate to investigate.  Moreover these 2 markers must have more data in basic research. Authors had better add these information. 

Author Response

Dear Journal of Clinical Medicine Editor and Reviewers:

We appreciate the time and effort that you and the Reviewers have dedicated to providing valuable feedback on our manuscript (jcm-1247148).

We are also glad to hear the positive comments of Reviewers, who remark that the study has been well performed, clearly presented and the manuscript is well written.

We have addressed all the insightful comments and suggestions made by Referees which undoubtedly will contribute to a better reading and understanding of the results presented. A point-by-point response to the Reviewer´s suggestions follows. Text changes have been highlighted in yellow in the “Re-uploaded Manuscript”.

Looking forward to hearing from you in due time regarding our submission.

Sincerely,

Dr. Andrés Da Silva Candal

Clinical Neurosciences Research Laboratory

Hospital Clínico Universitario

Santiago de Compostela (Spain)

E-mail: andres.alexander.da.silva.candal@sergas.es

Reviewer 1

This paper reports that glutamate/IL-6 index on admission >5 is associated with latency time of < 121 minutes from onset of symptoms.

This study included sufficient number to analyze (4775 ischemic stroke patient), but I think there are some points that authors had better revise. At first, a lot of research papers about biomarkers of stroke onset has been reported. Authors had better write these data more in detail. 

We thank the reviewer for his feedback on this aspect, to facilitate the general overview of biomarkers in stroke we have added the following references in the introduction section:

  • Jickling GC, Sharp FR. Blood biomarkers of ischemic stroke. Neurotherapeutics. 2011;8:349-360
  • Dagonnier M, Donnan GA, Davis SM, Dewey HM, Howells DW. Acute stroke biomarkers: Are we there yet? Frontiers in Neurology. 2021;12
  • Jickling GC, Sharp FR. Blood biomarkers of ischemic stroke. Neurotherapeutics. 2011;8:349-360
  • Kamtchum-Tatuene J, Jickling GC. Blood biomarkers for stroke diagnosis and management. NeuroMolecular Medicine. 2019;21:344-368
  • Laskowitz DT, Kasner SE, Saver J, Remmel KS, Jauch EC, Group BS. Clinical usefulness of a biomarker-based diagnostic test for acute stroke: The biomarker rapid assessment in ischemic injury (brain) study. Stroke. 2009;40:77-85
  • Glushakova OY, Glushakov AV, Miller ER, Valadka AB, Hayes RL. Biomarkers for acute diagnosis and management of stroke in neurointensive care units. Brain Circ. 2016;2:28-47

Secondly, authors selected two inflammation biomarkers, glutamate and IL-6. Authors had better write why they chose only these 2 markers in detail because there must be other inflammation biomarkers to become candidate to investigate.  Moreover these 2 markers must have more data in basic research. Authors had better add these information.

We thank the reviewer for his clarification in this regard and we agree with the need to clarify this point for a better understanding.

On the one hand, glutamate is fully accepted as an excitotoxicity marker in cerebral ischemia and its high levels are related to a worse prognosis. Likewise, IL-6 is considered a specific pro-inflammatory and prognostic marker in cerebral ischemia which validated both as specific markers of the processes described in our work. In any case, we fully agree with the reviewer that the use of other markers could be ideal, potentially providing more information and results; however, the retrospective nature of the study limits the use of new markers to those previously analyzed in these samples. The positive results of this work favor the development of new prospective studies that expand the range of markers used, for which we thank the reviewer for his suggestion.

Also, as suggested by the reviewer for a better understanding of the markers used, we have modified the introduction in the main manuscript for a better description and added the pertinent bibliographic references as follows:

“In this study we analyzed two time-dependent processes of ischemic stroke. Firstly, we examined glutamate levels as a marker of excitotoxicity25, a molecule involved in secondary neuronal death during the acute phase and related to the severity of injury and patients outcome26. Secondly, we analyzed interleukin-6 (IL-6)27, 28 a pro-inflammatory molecule implicated in early stages of stroke29, 30 and which has also been shown to be involved in different pathways that increase the damage severity 31 and patient outcome30. We carried out a combined analysis of this two temporal markers in order to develop an index to determine stroke symptoms onset.”

Note: we have included the responses to the reviewers in PDF format for a better and clearer visualization.

Reviewer 2 Report

This is an interesting study regarding the temporal trend of biomarkers (glutamate as a surrogate for excitoxicity and IL-6 for inflammation) during the acute phase of ischemic stroke. Acute ischemic stroke patients were divided in four quartiles. The index glutamate/IL-6 ratio showed a strong correlation with the earlier presentation of patients after symptoms onset. Regarding the population of the study there seems to be an ambiguity. Quartile 1 and Q2 have 1103 pts each one, Q3 has 1117 pts and Q4 1085. However, the total number of glutamate levels was 1568, which means that at total 1568 pts were analyzed. Similarly for IL-6 (n=1430). So there is a discrepancy between the 4775 patients in total and 1568 for whom there were data regarding glutamate and 1430 regarding IL-6.

  1. Is further analysis restricted only to patients for whom there were data regarding the indexes or to the whole cohort? I suggest limiting the analysis to only those patients analyzed. A flowchart of the study would be also helpful in order to follow the analysis and highlight any dropouts.
  2. Please also provide the number of wake-up strokes for each quartile. I suggest also providing a subanalysis (or sensitivity analysis) by excluding these patients from the analysis.Given that the purpose of the study is the temporal trend of the indexes, time from last seen well which is a convention from the first rtPA studies, might not be appropriate for this analysis.

3.Disussion line 211. Please erase the phrase “hemorrhagic” given that the study is confined to ischemic strokes exclusively.

  1. Conclusions line278 “short latency times 0-133” may be a typo (Q1 is 0-121min)?

Author Response

Dear Journal of Clinical Medicine Editor and Reviewers:

We appreciate the time and effort that you and the Reviewers have dedicated to providing valuable feedback on our manuscript (jcm-1247148).

We are also glad to hear the positive comments of Reviewers, who remark that the study has been well performed, clearly presented and the manuscript is well written.

We have addressed all the insightful comments and suggestions made by Referees which undoubtedly will contribute to a better reading and understanding of the results presented. A point-by-point response to the Reviewer´s suggestions follows. Text changes have been highlighted in yellow in the “Re-uploaded Manuscript”.

Looking forward to hearing from you in due time regarding our submission.

Sincerely,

Dr. Andrés Da Silva Candal

Clinical Neurosciences Research Laboratory

Hospital Clínico Universitario

Santiago de Compostela (Spain)

E-mail: andres.alexander.da.silva.candal@sergas.es

Reviewer 2

This is an interesting study regarding the temporal trend of biomarkers (glutamate as a surrogate for excitoxicity and IL-6 for inflammation) during the acute phase of ischemic stroke. Acute ischemic stroke patients were divided in four quartiles. The index glutamate/IL-6 ratio showed a strong correlation with the earlier presentation of patients after symptoms onset.

- Regarding the population of the study there seems to be an ambiguity. Quartile 1 and Q2 have 1103 pts each one, Q3 has 1117 pts and Q4 1085. However, the total number of glutamate levels was 1568, which means that at total 1568 pts were analyzed. Similarly for IL-6 (n=1430). So there is a discrepancy between the 4775 patients in total and 1568 for whom there were data regarding glutamate and 1430 regarding IL-6. Is further analysis restricted only to patients for whom there were data regarding the indexes or to the whole cohort? I suggest limiting the analysis to only those patients analyzed.

We agree with the reviewer with the possible misunderstanding in the manuscript, because the total population of the study and the number of glutamate and IL-6 analyses do not match. The clinical analyses as well as demographic aspects (Tables 1, 2 and 3) were analyzed in the total population stipulated in the study, however the glutamate and IL-6 molecules were not previously analyzed in the whole population because the analysis of these two molecules does not correspond to the usual clinical practice and they were previously analyzed for different purposes, so due to the retrospective nature of this work prevents the analysis of more patients than those included to date. In order to avoid possible misinterpretations we stated so in the main manuscript, in the Analytical Measurements sub section, Methods section:

“To calculate the Glut/IL-6 ratio, serum glutamate values were divided by serum IL-6 values. Of the total series, the number of patients with glutamate levels analyzed was 1568. And 1430 patients were analyzed for the IL-6 levels. A total of 1221 patients had both glutamate and IL-6 levels. The univariate, multivariate analyses and the Glut/IL-6 ratio described in the results were performed on these patients.”

We have also included a flow chart in which the total number of patients analyzed and how many of them have the Glu/IL-6 ratio analyzed for a better understanding of the work.

We thank the reviewer for the clarification, the potential good results suggest that a broader prospective study should be carried out to confirm by one side these results as well as to add new and clearer ones.

- A flowchart of the study would be also helpful in order to follow the analysis and highlight any dropouts

We have included a flowchart of the study design to facilitate the reader's understanding as suggested by the reviewer (attached below) and has been included as Supplemental Figure 1.

Supplemental Figure 1. Study design flowchart

- Please also provide the number of wake-up strokes for each quartile.

To facilitate the understanding of the results, we have included the number of awakening strokes in the suggested flowchart (included previously), which has also been included as Supplemental Figure 1.

- I suggest also providing a subanalysis (or sensitivity analysis) by excluding these patients from the analysis. Given that the purpose of the study is the temporal trend of the indexes, time from last seen well which is a convention from the first rtPA studies, might not be appropriate for this analysis.

We agree with the reviewer about this issue and because of this in the manuscript we attached as Supplemental Table 3 an analysis of the markers without including the awakening strokes that showed similar results (attached below).

Supplemental Table 3. Glutamate and interleukin-6 univariate analysis in stroke patients (excluding awaking strokes) subdivided by difference latency times.

Q1

Q2

Q3

Q4

p

Glutamate, mM/mL

302.7 ± 88,6

128.2±53.9

60.6±64.8

134.5±90.2

<0.0001

IL6, pg/mL

18.1±12.4

29.6±13.3

31.8±13.5

35.4±11.4

<0.0001

- Disussion line 211. Please erase the phrase “hemorrhagic” given that the study is confined to ischemic strokes exclusively.

We regret the error and it has been corrected in the main manuscript, thanks to the reviewer for the clarification.

- Conclusions line278 “short latency times 0-133” may be a typo (Q1 is 0-121min)?

Thanks to the reviewer for noticing the error, it has been modified to include the correct latency time of “0-121 minutes”.

Note: We have included the responses to the reviewers in PDF format for a better and clearer visualization.

Round 2

Reviewer 1 Report

This paper reports that glutamate/IL-6 index on admission >5 is associated with latency time of < 121 minutes from onset of symptoms. This study included sufficient number to analyze (4775 ischemic stroke patient). Authors could answer my questions well.

Reviewer 2 Report

The authors have adequately responded to the issues raised in the previous round of review. No further comments exist on my behalf.